# Engineering 3D Graphene-Based Materials: State of the Art and Perspectives

**DOI:** 10.3390/molecules25020339

**Published:** 2020-01-14

**Authors:** Luca Bellucci, Valentina Tozzini

**Affiliations:** Istituto Nanoscienze–CNR and NEST-Scuola Normale Superiore, Piazza San Silvestro 12, 56127 Pisa, Italy; luca.bellucci@nano.cnr.it

**Keywords:** graphene-based materials, nanoporous graphene, epitaxial graphene, molecular modeling

## Abstract

Graphene is the prototype of two-dimensional (2D) materials, whose main feature is the extremely large surface-to-mass ratio. This property is interesting for a series of applications that involve interactions between particles and surfaces, such as, for instance, gas, fluid or charge storage, catalysis, and filtering. However, for most of these, a volumetric extension is needed, while preserving the large exposed surface. This proved to be rather a hard task, especially when specific structural features are also required (e.g., porosity or density given). Here we review the recent experimental realizations and theoretical/simulation studies of 3D materials based on graphene. Two main synthesis routes area available, both of which currently use (reduced) graphene oxide flakes as precursors. The first involves mixing and interlacing the flakes through various treatments (suspension, dehydration, reduction, activation, and others), leading to disordered nanoporous materials whose structure can be characterized *a posteriori*, but is difficult to control. With the aim of achieving a better control, a second path involves the functionalization of the flakes with pillars molecules, bringing a new class of materials with structure partially controlled by the size, shape, and chemical-physical properties of the pillars. We finally outline the first steps on a possible third road, which involves the construction of pillared multi-layers using epitaxial regularly nano-patterned graphene as precursor. While presenting a number of further difficulties, in principle this strategy would allow a complete control on the structural characteristics of the final 3D architecture.

## 1. Introduction

Since the experimental confirmation of its existence [1], graphene has raised great expectations because of its exceptional properties, stemming from a fortunate combination of the electronic structure of carbon, the symmetry of its lattice, and its two-dimensional (2D) nature [2]. Besides the large charge carriers’ mobility and the wide-band optical response, graphene displays extremely large resistance to tensile strain associated to a very low bending rigidity [3], leading among other things to the emergence of low energy transverse phonons [4] and ripples [5]. These properties associated to the low weight have triggered the proposal of a plethora of possible applications [6,7,8,9,10].

With little exceptions, however, these require some sort of manipulation of the sheet: in general nano-electronics requires doping to increase the density of states at the Fermi energy or to open a gap, which can be achieved by chemical substitutions [11], introduction of adatoms [12] or defects [13,14] or structure modulation [15,16]; for photovoltaics [9] different functionalization are required, depending on the specific use proposed (anode, cathode or photoactive layer [17]). Catalysis or environmental applications, such as water filtering, generally require sheet alteration, such as perforations of tailored size [18]. Recently, a brand new branch of investigation has stemmed from graphene in-plane large mechanical strength and elasticity [3], coupled to out of plane flexibility [5]: controlled local strain would create pseudo-magnetic fields [19], besides band-gap opening and other specific electronic structure modifications [20,21] with interesting applications in nano-electronics and photonics; both in-plane (strain [22]) and out-of-plane (rippling [23]) mechanical alterations were shown to locally change chemical reactivity opening the way to controlled chemical nano-patterning [24,25]. Interestingly, all of the suggested modifications correspond to controlled disruption of the perfect symmetry of the crystal in a different way, which leads to viewing grapheme–rather than a single material–as a sort of morphable platform to build a class of slightly different materials suitable to specific purposes [26,27].

In addition to the modification of the layer, a wide range of uses needs its volumetric extension, with the requirement, however, that the 2D properties are preserved as far as possible. This is the case in applications involving storage: fuel-gas storage (e.g., H_2_), or electro-chemical energy storage (supercapacitors or batteries [28]), require a large exposed surface per unit mass (or Specific Surface Area, SSA) to achieve a large gravimetric capacity (GC) [29], and the intrinsic capability of adsorbing specific substances (gas or electrolytes). Similar requirements are needed in catalysis applications [30]. Clearly, in this case the light weight of carbon and its intrinsic two-dimensionality are crucial, electric conductance is also needed in supercaps and batteries, while electronic properties can be an important added value. Finally, a number of applications related to coating can be considered as in between superficial and volumetric ones. In these, graphene-based materials must be deposited on a given surface in thin layers–but macroscopic on the atomic scale–to several purposes: protect from atmospheric agents [31], make it conductive [32] or hydrophobic [33], yet maintaining elasticity and resistance.

Indeed, preserving the needed properties and possibly enhancing or tailoring them in the 2D to 3D passage has turned out extremely complex. Up to now, two main routes were considered, both using graphene flakes as precursors. In the first, these are created by graphite exfoliation (usually after oxidation) and suspended in various solvents, resulting in a mixture of flakes with randomly distributed sizes and shapes; upon dehydration, they form 3D scaffolds with random structure and porosity [34]. These techniques, described in the next section, have the advantage of producing in cheap and scalable way a range of different 3D graphene-based nanoporous materials (GNM). The disadvantage is the high level of disorder, and the poor capability of controlling structural and mechanical properties, which are usually characterized *a posteriori*.

Building multi-layered structures separated by molecular “pillars” is considered an alternative to control the properties of the final 3D construct: theoretically, porosity and density in such structures are determined by the size and concentration of pillars, allowing the possibility of engineering the 3D structure via the pillar molecule design. Up to now, this route, has been followed using organic molecules as pillars [35,36] coupled to suspended flakes, with encouraging but still not optimal results due to the difficulty of controlling the location of pillars on the randomly shaped flakes. The latest advances in this field are reviewed in Section 3.

Clearly, the optimal route should involve the control of the pillars positioning at the nano-scale on the precursor sheets, i.e., the combination of controlled chemical nano-patterning with the possibility of stacking the patterned multilayers in a controlled way. In Section 4 we illustrate the perspective to reach these objectives using the epitaxial graphene as precursor. A summary and conclusions follow in the last section.

## 2. Graphene-Based Nano-Porous Materials: Production and Computer Modeling

GNM are part of a broader class, the nano-porous carbons, which comprises activated carbons, carbide derived carbons, nanofoams and nanotubes, among others. While the synthesis of scaffolds with micrometer porosity has reached quite a high level of maturity thanks to the use of nano-to-micro particles as templates [37], strictly nano-porous GNMs are more difficult to produce with controlled structural characteristics. They are generally obtained with top-down techniques, using as precursors suspension of flakes. Flakes obtained from direct exfoliation of graphite (e.g., by liquid phase exfoliation [6,7]) have more regular structure and better conductive properties, and are therefore more suitable for electronics applications, but are also more expensive and difficult to obtain and handle. Therefore, for the 3D scaffold building, usually, the process starts from the oxidation of graphite to graphite oxide, e.g., by Hummer’s method [38], followed by exfoliation-reduction either thermally [39], leading to Thermal Exfoliate Graphite Oxide (TEGO), or using microwaves, leading to Microwave Exfoliate Graphite Oxide (MEGO) [40,41], resulting in materials with SSA usually not exceeding 800 m^2^/gr [42,43]. Samples can be subject to additional treatments, such as further reduction, or chemical activation (e.g., with KOH), which modify the edges with the result of increasing the porosity to specific pore volume (SPV) greater than 2 cm^3^/g [30,44] and improving the SSA up or exceeding the graphene limit (2630 m^2^/g). The result are 3D structures with randomly distributed sp^2^ areas interconnected to form a tangled scaffold with pores of size ranging in the nanometer scale (Figure 1). Overall, these materials display SSA values between 500 and 3500 m^2^/g, maintaining good electrical conductivity, high mechanical strength and chemical stability [45,46]. The performances as gas absorbers are basically proportional to the SSA, reaching an excess H_2_ adsorption of 7% at 77 K [44]. The actual structural features, measured by SSA and PSV and some other additional parameters, such as the pore size distribution (PSD) and the mass density *ρ* [47] (see Table 1), depend on all the phases of the production: the exfoliation process, determining the size and shape distribution of the flakes, the reduction, influencing the intrinsic perforation and defects of the flakes, and the activation, modifying the porosity and surfaces. Consequently, the gas adsorption could in principle be tuned provided a full control of the production process is possible.

The structure control is even more crucial when GNM are proposed as storage mean in electric or electrochemical form. Being a conductor with large surface, graphene could be used as a capacitor, whose capacitance can be largely increased adsorbing electrolytes, potentially making it a super-cap [40]. To this aim, besides the already mentioned SSA directly related to capacitance and improved by activation [48], also the intrinsic capability of adsorbing electrolytes or ionic liquids becomes a key feature [49]. Therefore, though the capacitance is generally inversely proportional to porosity, the pore sizes must also be optimized based on the size of the ionic species [50,51]. A fine tuning of the porosity can also produce ion desolvation and the consequent increase of efficiency via a pseudo-capacitance effect [52]. Similar properties are required to develop materials suitable for batteries. In particular, electric conductivity and chemical stability, besides porosity are the main requisites for the electrodes for lithium-based batteries [53,54]. Finally, GNM are attractive also as gas sorters or filters for environmental applications, e.g., water or air purification and CO_2_ sequestration [55].

In summary, the need for large GC for gas or electrolytes adsorption calls for large SSA and low *ρ* [10,55], undermining the structural stability. On the other hand, volumetric capacity (VC) increases with *ρ*, and the pore size must be tuned to the adsorbed fluid [26]. Clearly, the capability of finely controlling the structural parameters has a main role [56]. This task is not only difficult, but also somehow ill-defined, since the experimental structural determinations of GNM are limited to the measurement of pair distribution functions (PDF) and average pore size or at most the PSD. Computer modeling and simulations have been called into play to compensate for the lack of detailed knowledge. However, the intrinsic disorder leaves quite a large amount of under-determinacy for model building. As a consequence, models including a degree of approximation or idealization are often used. The “perforated graphene” models [44] uses flat flakes not reconstructed at the edges and/or with regularly spaced pores [57]. Other studies are even simpler, including either the ideal slit-pore geometry [58,59,60] or defected [61]/rippled [62] multilayers. Finally, a number of models is based on periodic 3D structures, such as the open-carbon-frameworks [63,64] or the carbon honeycomb [65].

While regular structures cannot allow a comprehensive throughput screening of the whole structural diversity landscape of GNM, clearly, the major issue in building realistic models for nanoporous scaffolds is their intrinsic disorder, difficult to include and needing large model systems to mild the effects of the boundaries or of the superimposed artificial periodicity of the model super cell. A number of computational approaches to generate disordered GNM model systems were adopted, differing in the description of the interactions between carbon atoms [66], and in the technique used to sample the structural parameters space. In the molecular dynamics (MD)-based techniques, atomistic empirical force fields (FF) are used to handle the interactions. These must be able to describe the different possible hybridization states of carbon based on the bond-order evaluated “on the fly” [67,68] and/or the formation/dissociation of the different kinds of C-C bonds [69,70]. The model generation can then proceed ‘’bottom up’’, starting from a random distribution of carbon atoms in gas phase, which are subsequently subject to molecular dynamics simulated annealing cycles (heating up to 10^4^ K and slow quenching [71]). Different structural morphologies can be obtained by changing the annealing conditions [72] (temperature, pressure [73] or density [74]), which is the simulation equivalent of changing the experimental conditions of production. This “from scratch” procedure is very computationally expensive, limiting the size of the model system to tens of nm, and preventing an extensive exploration of the structural parameters space and–consequently–a fine control over the resulting structures. A completely different point of view is taken in reverse Monte Carlo methods, where the atoms configurations are generated randomly and optimized until the simulated PDF matches the experimental one. In principle, the bare version of this method returns the best approximation of the inter-atomic interaction with a two-body potential as a side result, and structures compatible with it [75]. However, the nature of the C-C interaction is intrinsically many-body, therefore further restrains (geometric or energetic) are needed during the procedure [76]. This method is less expensive and can then generate larger model systems, giving good results in the meso-scale, but needs accurate structural determinations as input, which necessarily introduce an experimental bias.

To the aim of combining a modest computational effort with realism of the final model, a good strategy is using as precursors already formed graphene portions [77,78] instead of atoms. On this road, a step forward was recently done using a model building algorithm that mimics the real synthesis [55,79]. The starting point is a mixture of flakes with size and shape distributed according to the experimentally known composition of the suspension. These are mixed to reproduce the real density and allow intersections. These, the edges and the perforations are then optimized using bond order or reactive FFs, and possibly functionalized with other species, mimicking the various experimental treatments. The system is finally refined by thermalizing MD cycles. The results realistically match the PDF and can be controlled by the starting concentration/size distribution/perforation of the flakes. Using already formed flakes as precursors, not only leads to more realistic structures, but also limits the computational effort allowing the generation and extensive study of large model systems. An overview of the available disordered GNM materials, models and their characteristics is reported in Figure 1 and Table 1.

## 3. Pillared Materials: State of the Art and Open Problems

In disordered nanoporous materials the porosity of the final structures depends on how the flakes interlace during all the phases of the preparation, which introduces a high degree of disorder and stochasticity. In order to reduce this issue and improve the control over the outcome, the idea rose of synthesizing layered structures separated by molecular pillars, i.e., organic molecules suitably designed with given lengths, rigidity and possibly other physic-chemical properties. The size of the pillars determines the inter-layer spacing, and controls the average pore size, together with the relative distance of the pillars on the sheets. The first realization of such structures traces back to almost two decades ago when, inspired by metal-organic frameworks chemistry, layered structures separated by diboronic acid molecules were first proposed [81]. The pillars adhesion exploited the reactivity of hydroxy groups of graphene oxide (GO) with the acidic groups, leading to GO frameworks (GOF). These were subsequently characterized via Xray diffraction and neutron scattering, and tested through their H_2_ adsorption capability, whose low value indicates rather a small SSA value (hundreds of m^2^/g). The synthesis procedure was recently optimized [34], obtaining values of SSA up to ~600 m^2^/g and pores size up to 2 nm. At the same time, it was shown that in some cases, in polar solvents these material exhibits reversible swelling, posing doubts on the complete covalent nature of the layer linkage [35]. Using as pillars di-amine of different lengths [82] resulted in materials with tuneable interlayer distance in the range 0.8–1.1 nm, generally hydrophilic and insulating. In fact, ab initio calculations with simplified model systems demonstrated that achieving electric conductivity in these materials is not easy, due to the rupture of the aromaticity at the linkage sites [80].

The optimization of these materials depends on their use: for electric energy storage, both a finely tuned pore size and the conductivity are important. Therefore different synthesis routes were explored, involving reduction of GO, either *ex post* [83] or directly starting from reduced GO (rGO); in the latter case, most of the proposed reaction exploit the chemistry of diazonium salts radicals, selectively reacting with the defective sites of rGO [84,85,86]. Among the best performances in terms of SSA where obtained with a two-step procedure: the rGO was first functionalized with benzoic acid [85], obtaining a layered material with good porosity, but scarce conductivity. Polyaniline was subsequently synthesised in situ obtaining a composite material with larger average inter-layer distance though smaller average pore size, and with improved electric conductance. Alternative routes to tailor inter-layer distance and porosity involve cross-linking by aryl-aryl reaction of rGO functionalized with iodo-phenyls [86] or Zn+ coordination of rGO functionalized with azobenzoic acid [87]. A summary of the recent literature on experimental and theoretical structural determinations of these materials is reported in Table 2. The main structural characteristics are also reported in Figure 1.

Although steps forward have been done in the control of the functionalization, the performances of these materials are not better than those of disordered GNM: SSA is at best several hundreds, far below the theoretical limits and below the simulation predictions. In fact, both the carbon-only model systems including nanotube-pillars [88,89] and the molecular-pillared model systems [81,90], display, in simulations, GD uptake (and SSA) 5–10 times larger than the measured ones, besides the theoretical capability of efficient gas sorting [91], desalinization [92], and interesting mechanical properties [93]. Although the origin of the theory-experiment discrepancy is not clear, it was shown that ideal structures with nearly flat sheets and regularly spaced pillars display a better performances in simulations [84], and that the adsorption performances depend on the fine tuning of the pillars distance, which must be large enough to allow the molecules access and hosting in a layer on the surface, but not too large, in order to maximize the GD. Therefore, a regular and controlled patterning seems the key for obtaining highly performing pillared materials.

## 4. Multilayers from Epitaxy: A Perspective

The reason why the pillars distribution is poorly controllable is encoded in the use of GO (or rGO) flakes as precursors: the covalent bonding of pillars or anchors exploits the presence of epoxy/hydroxy groups or defects, which are reactive sites [94]. However, these are randomly distributed, and their concentration is variable, and not easily tailorable [95]. In addition, flakes edges are also very reactive, attracting a relatively large number of functional groups, which introduces further disorder in the structure. Finally, the environmental conditions that promote the reaction (temperature, solvent, etc.) can favor aggregation in an almost unpredictable way. From this point of view, using epitaxial graphene as precursor would in principle bring some advantage, mainly related to the regularity of the material and to its laying on an extended solid support. In fact, this would allow a direct control over functionalization and check of results e.g., with atomic resolved microscopy techniques. One popular technique to produce supported graphene is chemical vapor deposition of carbon-rich compounds over metal substrates (after their cracking) [96]. Alternatively, one can use carbon rich substrates, such as SiC, and let the carbon layers reconstruct in the honeycomb lattice by selective evaporation of Si from surfaces with specific symmetries [97]. In both cases, one obtains macroscopic almost defectless single layers. In general, perfect graphene is poorly reactive, because of its fully delocalized stable sp^2^ electronic system. Clearly, reactivity can be brought back by reintroducing defects, e.g., by nitrogen sputtering [98], which creates either substitutional or structural defects, proven to act as seeds for adhesion of metal clusters or hydrogen [99]. However, these defects are introduced randomly, pushing back to the same problems as in GO flakes.

Indeed, specific kinds of epitaxial graphene offer different possibilities, which exploit the interaction with the substrate. For instance, radicals of diazonium salts are able to attach to sp^2^ sites but manifests a preference for graphene on hydrophilic substrates [100], due to charge accumulation effects. A similar effect is observed for graphene on metals such as iridium or ruthenium [101,102], where, in addition, a spatially modulated reactivity is created following the nano-metric moiré pattern of corrugation. This open the road to substrate driven functionalization, with the possibility of creating chemical nano-patterns following the symmetry of the moiré superlattice. Similar effects were obtained by intercalating metal clusters in between graphene and an insulating substrate [103], where the preferential adhesion of the radicals was observed in proximity of the metal cluster. The enhancement of reactivity (towards aryl radicals) is also observed on non-metallic substrates, such as patterned SiO_2_ [104] and on the protruding areas of the natural moiré corrugation lattice of monolayer graphene on SiC (towards atomic hydrogen [26]). In these cases, it is attributed to the curvature [24]. In fact, both rippling and strain [105] produce charge inhomogeneities. Therefore, supported graphenes with moiré patterns are very promising materials for substrate driven regular nano-patterning.

We now focus on graphene on SiC, to further explore this concept. It is important to observe that graphene on SiC is not a single material but includes different types of 2D carbons [106] that can be obtained with different procedures (see Figure 2a). Upon Si evaporation from the Si-rich surfaces with hexagonal symmetry, excess carbon produces in the first instance a hexagonal carbon buffer layer (BL) [107], covalently bound to the substrate, and partially sp^3^ hybridized. The bonds and corrugations follow a moiré pattern, due to the mismatch of the two lattices, displaying a hexagonal super-lattice of ~3.2 nm side, made of sharp crests and peaks with sp^3^-like pyramidal configuration [15]. Fully sp^2^ graphene can be obtained continuing evaporation: another BL forms under the first one, which is detached and becomes the so-called Mono-Layer graphene (MLG), characterised by a corrugation pattern with the same symmetry as BL, though smother [108]. Alternatively, the BL can be detached by intercalating H [108] or metals [109], obtaining the Quasi-Free-Standing Monolayer graphene (QFMLG) [110]. This is ideally flat, but displays in reality localized concavities, occupying the sites of a lattice roughly corresponding to 6 × 6 of SiC [111] with ~1.8 nm side, which were associated with vacancies of H in the intercalating layer [112]. The electronic structure is strongly affected by these defects, since Si dangling bonds produce electronic states localized near the Fermi level [113].

While all of the different carbon layers on SiC display charge inhomogeneities following a regular nano-pattern induced by the interaction with the substrate, either mediated by the hybridization, by the corrugation or by the vacancies in the intercalation coverage, only the GML was tested on its reactivity, showing selective H adhesion on the crests [26]. On the other hand, the localised electronic states forming on the QFMLG in corresponding of H-vacancies have various sizes and shapes, depending on the number and relative location of vacant sites and their energy is organized in groups of levels near the Fermi energy [113], indicating a possible propensity to electrophile attack. Even more interesting from the functionalization is the buffer layer (Figure 2b) since it displays the strongest deviation from graphene symmetry and the sharper definition of the moiré pattern [15]. Specifically, the sp^3^ cusps at the vertices of the moiré super-lattice are likely to be highly reactive sites in general, not only towards radicals, but possibly also towards e.g., dissociative chemisorption of H_2_. Conversely, the protruding crests, organized in diene like structures, and the intruding areas, organized in “benzene-like” rings [15] are likely to be attractive for cyclo-addition reactions [114], leading to a spatially complementary selectivity.

Clearly, the BL functionalization should be viewed as the first, yet fundamental, step of a procedure involving the multilayer formation (see Figure 2c): once the molecular anchors are attached, pillars of different length can be added exploiting, e.g., solvothermal de-hydration reactions; subsequently the layer should be exposed to similarly functionalised layers (previously detached by the substrate by intercalation) which have to be stacked and cross-linked. These steps are also taken in the already realized synthesis of pillared materials from GO or rGO flakes [90,91,92,93,94,95,96,97,98,99]. However, using regularly patterned precursors would offer two unique advantages: first, the space matching of cross-linking groups can potentially trigger the self-assembly of the sheets, greatly impring the efficiency of the process, and second, the final result would be a structure with pillars at controlled distance in the range of 2–3 nm. This, together with the inter-layer distance controlled by the pillar length, will result in a structure with pre-determined porosity. Clearly, exploring experimentally this strategy would benefit of preliminary computer simulations, which are currently work in progress.

## 5. Summary, Conclusions and Possible Developments

In summary, we have reported three possible routes to produce graphene-based materials with porosity on the nano-scale, ordered by increasing capability of control and tailoring of the final structure. The first class produces the disordered nanoporous scaffolds from GO or rGO flakes. These can reach large values of SSA and are, up to now, the most interesting for gas storage. However, controlling their final structure is not straightforward, because of the disordered structure of the precursors and of the stochastic nature of their combination during the production procedure. With the aim of controlling at least a part of the variables determining the porosity, the second strategy introduces on the flakes pillars molecules with pre-determined lengths and shapes. This produces a class of materials with average pore sizes at the nano-metric scale, matching with the size of electrolytes and therefore suitable for the use in electric and electro-chemical storage. However, the average value of the SSA of these materials is rather low, and the poor control over the distribution of the pillars on the sheet introduces disorder, preventing a full optimization, not only for supercapacitors and batteries, but also in catalysis and filtering applications.

A third route is currently in its infancy, which would provide a full control over the distribution and location of the pillars. This considers as precursors epitaxial graphene and exploits the electronic inhomogeneities of the sheet produced by the interaction with the substrate, typically following a nano-metric moiré pattern, for the controlled chemical functionalization. Although the first timid steps (selective functionalization with atoms or small molecules) were demonstrated, the way is long towards the production of multi-layers.

The support of computer modeling and simulations is essential in all cases: in the case of disordered scaffolds, the main issue is to create realistic models and to understand the relationship between production procedure and final structure, and between the latter and the adsorption performances; for the pillared (r) GO materials, the challenge is to control the concentration and location of the pillars and predict the properties as a function of the used pillar. Most of all, computer modeling will be of outmost importance in the pillared multi-layers building from epitaxial graphene. In this case, the simulation of the pillaring, stacking and cross-linking would be essential to give indications for the experimental realization of the procedure. Though extremely challenging, this strategy might give a full control over all the structural features of the resulting structure, and–acting on the nature of pillars–might allow to create brand new materials with tailored and unprecedented properties, such as locally tuned elasticity or conductivity, reactivity to external fields, optical response, and others.

## Figures and Tables

**Figure 1 molecules-25-00339-f001:**
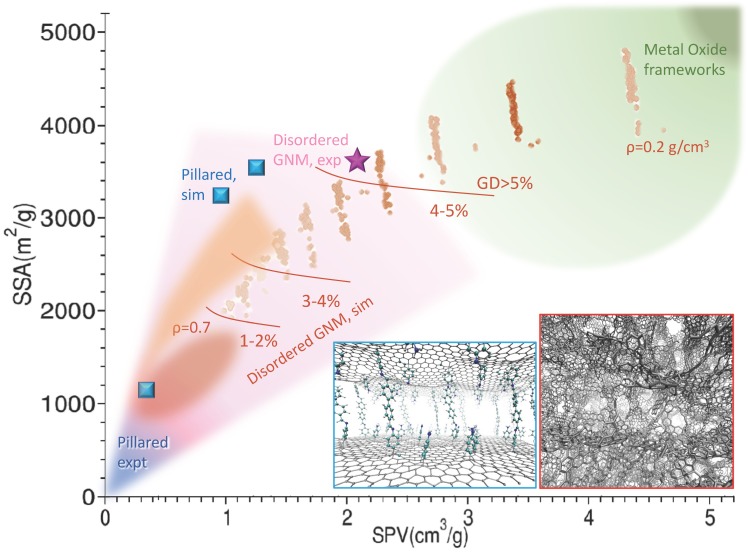
Structure diagram of graphene-based nanoporous materials (GNMs). The Specific Surface Area (SSA) vs specific pore volume (SPV) is reported for various experimental determinations and simulation studies. The blue shaded area encloses the experimental determinations of pillared structures, the one shaded in pink encloses the area spanned by experimental determinations of the disordered GNM scaffolds, both from various literature works cited in the text and in Table 2 (magenta star: Ref. [44]). The squared blue dots are simulations for pillared materials (Refs. [74,76,80]). Brown and reddish shades and dots are from simulations. The brown dots are preliminary from ref. [55], and roughly accumulates on lines at different decreasing density (smaller and larger simulated density are reported); red oval shade and orange shade are extracted and processed from ref. [74]. The brown lines separate areas at increasing excess GD evaluated at 77K. The region typically spanned by the Metal Oxide Frameworks is reported in green. Sample structures for the pillared (blue border) and disordered GNM (red border) are reported as insets.

**Figure 2 molecules-25-00339-f002:**
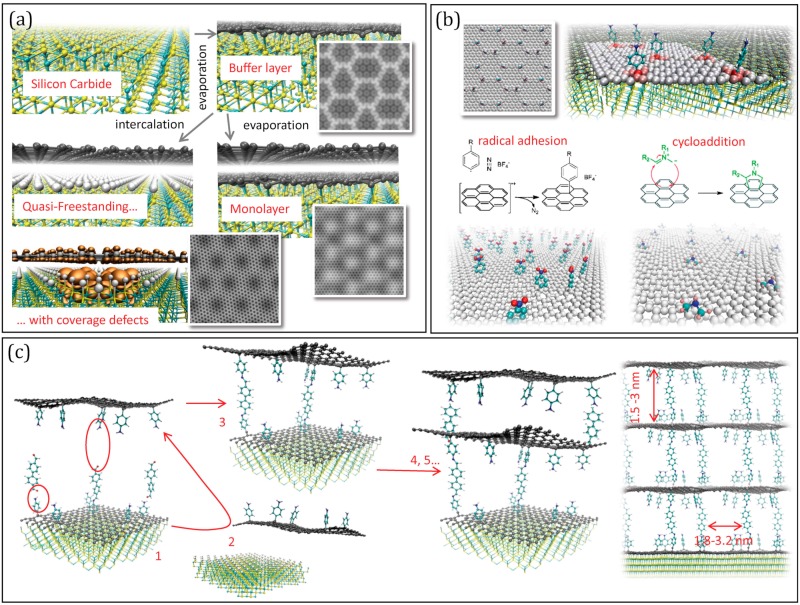
(**a**) Summary of the production of graphenes on SiC: the Buffer layer (BL by evaporation of Si), graphene monolayer (GML by subsequent evaporation) and quasi-free-standing monolayer (QFMLG, by intercalation of H or metal). The simulated Scanning Tunnel Microscopy images are reported for the BL, for the ML and for the QFSML with defects in the intercalation coverage layer. (**b**) Scheme of possible functionalization reactions exploiting the corrugation pattern of the BL. (**c**) Scheme of a possible strategy to build pillared multilayers: after pillaring (1), the cross-linking should occur with a previously detached functionalized sheet (2 to 3), and be re-iterated (4, 5…) to give a regular structure.

**Table 1 molecules-25-00339-t001:** Disordered GNM and their structural characteristics.

Precursor	Method/Treatment	SSA m^2^/g	PSV cm^3^/g or Avg Pore Size	Density cm^3^/g	H_2_ Uptake (% at 77K) or Capacitance (F/g)	Ref.
Graphite oxide	TEGO, TEGO + KOH	2300			5%	2015 [29]
Graphite oxide	TEGO + KOH	3300	2.2 (PSV)		7%	2015 [44]
Graphite oxide	TEGO + KOH	2900	1.4 (PSV)	~1	5.5%	2015 [78]
Slit pores	Modelling	5100	0.95 (PSV)	~1	6.5%	2015 [78]
Graphite	plasma-induced exfoliation	~800	~0.8 nm		2%	2016 [79]
Graphite-/diamond-like	Heating/Quenching MD simulations	600–3000	0–1.6 (PSV)	0.5–3.5		2017 [72]
activated carbon	Thermal treatment	2220	0.67 nm	1.95	5.5%	2015 [47]
Carbon atoms	Quench MD simulations	~1900	3–15 nm	~0.9	123 F/g	2019 [71]

**Table 2 molecules-25-00339-t002:** Pillared materials derived by Graphite Oxide (GO) or reduced GO (rGO) flakes and their structural characteristics.

Precursor	Pillars	Reaction/Method	SSA m^2^/g	Structural Features	H_2_ Uptake (% at 77K)	Ref.
GO	Diboronic acid	Solvothermal Acid+OH dehydration	~200	~11 Å interlayer spacing; pillars distance: 7–8 Å	1% experiment 5% simulation	2010 [80]
GO	Diboronic acid	Solvothermal	500–600	Interlayer: 8–15 (swelling) Pore size > 2 nm	~1.5%	2015 [34]
GO	“tetrapod” amine	Solvothermal	>660	Interlayer: 10–13 to ~16 Å (swelling)	~1.5%	2017 [35]
GO	Different types of diamine	Cross-linking, thermally promoted		Interlayer 8.5–11 Å Pillar dist ~10 Å		2019 [84]
GO reduced	1–6 diaminohexane	Cross-linking	150–200	Inter layer: 7.8 Å Pore size: 1 nm, 15 nm		2018 [85]
rGO	Aryl bis-diazionium salts (and variants)	Radical reaction	200–400	Interlayer: 5–10 Å inter-pillar ext: ~5 Å		2016 [86]
rGO	Benzoic acid, polyaniline	Polyaniline is grown on benzoic acid on flakes	330	Inter layer 1.5–2.5 nm Density 0.68 g/cm^3^ Pore size 0.8 nm		2015 [87]
rGO	4-iodophenyl diazionium salts	Aryl-aryl coupling reaction for cross-linking		Pore size 1–10 nm		2015 [88]
rGO	Azobenzoic acid-based ligands	Zn^2+^ coordination for cross-linking		inter-layer distance ~3 nm in the hydrogel		2012 [89]
graph	Diboronic acid variants	Density Functional Theory, Tight binding		Interlayer 1.1–2.2 nm inter pillar 3–5 Å	1.5%	2019 [84]
graph	nanotubes	Density Functional Theory, Grand Canonical MC		1.2 nm interlayer, 1.5 nm inter-pillar	6%	2017 [71,72]
GO, gr	Organic aromatic pillars	Reax FF Grand Canonical MC		Pore size 0.8,1,1.1 nm Inter-layer ~3 nm	~4%	2017 [73,74]

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
