# Peer review of "Engineering 3D Graphene-Based Materials: State of the Art and Perspectives"

_molecules, 2020, doi:10.3390/molecules25020339_

Round 1

Reviewer 1 Report

Graphene-based materials is one of the most promising topics in modern nanotechnologies.

The review is balanced enough. The cited studies are recent enough and represent the current knowledge on the topic. The personal opinions of the authors in the summary section represent informed and considered deductions. 

I know that the referee reports without constructive suggestions may be sometime considered as worthless and unhelpful, but this is not the case here. The manuscript under consideration is an example of the professionally made work -- made by professionals, made for professionals.

Recommendation: Accept.

Author Response

We thank the reviewer for her/his comments.

Reviewer 2 Report

The authors reported three strategies of producing graphene-based nanomaterials and compared them in the capability of control and tailoring. The potential advantages and challenges of using epitaxial graphene as precursors were discussed and summarized, which could guide more research work in substrate driven functionalization. Minor revision is suggested and a few comments in details:

In line 92, the authors use the statement “either…or…” to summarize the exfoliation-reduction strategies. Is there any reason to skip other methods that can also realize this step such as chemical exfoliation based on Hummer’s method? In line 99, the reference issue needs to be addressed. The font in Figure 2 is a bit messy and suggested to be modified. In figure 2c, the two arrows at the bottom are confusing. Which pattern is supposed to be the start of the route? Besides, in the pillaring procedure, how to control the length of pillars? Their lengths seem hard to control according to the scheme, which might lead the interlayer distance difficult to control. A lot of exfoliation methods have been developed to produce graphene by using graphite as the precursor instead of GO, to name some, gas exfoliation, redox exfoliation, and liquid-phase exfoliation. How would you compare these methods with the three strategies reported in the capability of control and tailoring?

Author Response

Reviewer 2
The authors reported three strategies of producing graphene-based nanomaterials and compared them in the capability of control and tailoring. The potential advantages and challenges of using epitaxial graphene as precursors were discussed and summarized, which could guide more research work in substrate driven functionalization.
We thank Reviewer 2 for this clear summary of our work. We answer to her/his comments in the following
1. Minor revision is suggested and a few comments in details:
In line 92, the authors use the statement “either…or…” to summarize the exfoliation-reduction strategies. Is there any reason to skip other methods that can also realize this step such as chemical exfoliation based on Hummer’s method?
We thank the reviewer for this comment, which indicates that our sentence was not completely clear. Hummer’s method is an oxidation procedure, and we did not intend at all to skip he oxidation step in the exfoliation procedure. In fact the complete sentence reads

“ Usually, the process starts from the oxidation of graphite to graphite oxide (GO), followed by exfoliation-reduction either thermally, […] or using microwaves…”.

In the logic of our writing, the chemical approach to oxidate the graphite is in fact a preliminary step and by exfoliation we intend the subsequent one in which GO flakes are actually detached, suspended etc. In one previous version of our manuscript there was an explicit mention to Hummer’s method, which we eliminated to simplify. We re-added in this version for clarity and now the sentence reads

“Usually, the process starts from the oxidation of graphite to graphite oxide, e.g. by Hummer’s method followed by exfoliation-reduction either thermally, […] or using microwaves…”

2. In line 99, the reference issue needs to be addressed.
Done

3. The font in Figure 2 is a bit messy and suggested to be modified. In figure 2c, the two arrows at the bottom are confusing. Which pattern is supposed to be the start of the route?
We uniformed the fonts of Fig 2. Fig 2 c is read from left to right, therefore the first figure to read is the one on the left, and then following the arrows. We tried to make it clearer with more readable arrows and adding numbers to follow the sequence. The caption has been changed accordingly.

4. Besides, in the pillaring procedure, how to control the length of pillars? Their lengths seem hard to control according to the scheme, which might lead the interlayer distance difficult to control.
This is indeed a crucial point. The use of pillars of different length and properties is precisely what makes the pillaring method interesting. Probably we did not stress enough the fact that the the length of pillars is controllable chemically: a large number of molecules with different length, rigidity and other tuned chemical-physical characteristics is available and could be used for the described purposes. This has been already done, as mentioned e.g. on lines 198-200 (ref 82). In order to make this concept clearer, we added a sentence on line 188-189 (added part reported in red here below)

“In order to reduce this issue and improve the control over the outcome, the idea rose of synthesizing layered structures separated by molecular pillars, i.e. organic molecules suitably designed with given lengths, rigidity and possibly other physic-chemical properties. “

5. A lot of exfoliation methods have been developed to produce graphene by using graphite as the precursor instead of GO, to name some, gas exfoliation, redox exfoliation, and liquid-phase exfoliation. How would you compare these methods with the three strategies reported in the capability of control and tailoring?

These methods are mostly aimed at producing graphene flakes or crystals with the best quality, i.e. with a few defect and high electron mobility. They are less used to produce 3D scaffolds or for pillaring, both because they are more expensive and less scalable, and because - maybe paradoxically - the lack of defects is a problem for functionalization, and realization of 3D structures. However, the reviewer is right, these methods deserve at least a mention. we added it on line ~90 (red part added)

“Flakes obtained from direct exfoliation of graphite (e.g. by liquid phase exfoliation6,7) have more regular structure and better conductive properties, and are therefore more suitable for electronics applications, but also more expensive and difficult to obtain and handle. Therefore, for the 3D scaffold building, usually, the process starts from …”

Reviewer 3 Report

The authors present and compare somehow in this review article two different ways to build 3D structures from graphene describing how can someone combine the properties of graphene such as conductivity, large surface area, chemical stability etc. This article is a significant effort to summarize achievements about graphene 3D materials and I suggest the acceptance for publication after some minor revisions that could help the authors to improve it. 

a) The terms graphene, graphene oxide, reduced grapene oxide must be checked again in teh manuscript in general. It must be clear what is really the material each time. For example

..in page 66 the phrase '' graphene (or reduced graphene oxide, GO) is not clear and should be reformed. I think GO is almost exclusively used in such procedures. The authors state also this in other points. 

..in page 67. ''graphite exfoliation'' combined with ''dehydration'', again it is not clear that here the authors are reffered to GO as ref 34 do.

Author Response

Reviewer 3
The authors present and compare somehow in this review article two different ways to build 3D structures from graphene describing how can someone combine the properties of graphene such as conductivity, large surface area, chemical stability etc. This article is a significant effort to summarize achievements about graphene 3D materials and I suggest the acceptance for publication after some minor revisions that could help the authors to improve it. 
We thank the reviewer for the useful summary of our work and for the words of appreciation. We addressed her/his issues as follows

a) The terms graphene, graphene oxide, reduced grapene oxide must be checked again in teh manuscript in general. It must be clear what is really the material each time. For example
..in page 66 the phrase '' graphene (or reduced graphene oxide, GO) is not clear and should be reformed. I think GO is almost exclusively used in such procedures. The authors state also this in other points. 
..in page 67. ''graphite exfoliation'' combined with ''dehydration'', again it is not clear that here the authors are reffered to GO as ref 34 do.

We thank the reviewer for these comments, they were very useful to solve ambiguous terminologies.

Concerning the sentence
“Up to now, two main routes were considered, both using graphene (or reduced graphene oxide, GO) flakes as precursors”

The reviewer is right, GO or rGO is generally used for that aim, as we clearly explain in the manuscript, in section 2. However, in the introduction, we intended to give only a generic idea.
On the other hand is also true that by “graphene flakes” one generally intends flakes derived from graphite, including GO flakes in solutions obtained after oxidation/exfoliation, but also for instance rGO flakes obtained after subsequent reduction.
Having this in mind, we changed the sentence as follows

“Up to now, two main routes were considered, both using graphene flakes as precursors. In the first, these are created by graphite exfoliation (usually after oxidation) and suspended in various solvents, resulting in a mixture of randomly distributed sizes and shapes; upon dehydration, they form 3D scaffolds with random structure and porosity.”

In the first part, graphene flake is used in the generic sense. In the second part it is specified that the flakes used are obtained after oxidation, and therefore are GO flakes.

All the details are then explained in section 2, with no ambiguity, we believe.

Additionally, we resolved ambiguities using GO only for graphene oxide and rGO for reduced GO. We made the needed changes accordingly

Reviewer 4 Report

The manuscript titled “Engineering 3D graphene based materials: state of 2 the art and perspectivesby Luca Bellucci et al reviewed the synthetic strategies of 3D graphene-based materials with porosity. Overall this is a high-quality manuscript. I recommend this manuscript publish in this Journal after English errors are corrected.

Author Response

We thank the reviewer for her/his comments. We performed an exhaustive grammar check, we hope that in the current version our manuscript is acceptable.